# Experimental Study on the Mix Proportion and Mechanical Properties of New Underwater Cementitious Filling Materials

**DOI:** 10.3390/ma15082938

**Published:** 2022-04-18

**Authors:** Yuan Mei, Dongbo Zhou, Rong Wang, Miaomiao Zhang, Wangyang Xu, Yan Zhang, Xin Ke

**Affiliations:** 1School of Civil Engineering, Xi’an University of Architecture & Technology, Xi’an 710055, China; meiyuan@xauat.edu.cn (Y.M.); wangrong97@xauat.edu.cn (R.W.); zhangmiaomiao@xauat.edu.cn (M.Z.); xuwangyang@xauat.edu.cn (W.X.); zhangyan0603@xauat.edu.cn (Y.Z.); kexin20222022@163.com (X.K.); 2Shaanxi Key Laboratory of Geotechnical and Underground Space Engineering, Xi’an 710055, China

**Keywords:** karst, NUC-FM, mix proportion, mechanical properties, microstructure

## Abstract

Considering that it is difficult for traditional materials to simultaneously meet the requirements for filling grouting of water-filled karst caves and subsequent shield tunneling, an environmentally friendly and controllable new underwater cementitious filling material (NUC-FM) is developed, with abandoned shield mud as the basic raw material. Through laboratory tests, the mechanical property parameters of NUC-FM are tested, and its micromechanism is analyzed. The research results show that there is excellent synergistic interactions among shield mud, cement, flocculant, fly ash and other raw materials. The NUC-FM grouting filling material with superior performance can be prepared when the water binder ratio is between 0.45 and 0.6 and the water consumption is between 270 and 310 kg/m^3^. It has the characteristics of non-dispersion underwater and moderate consolidated body strength. The compressive strength of the NUC-FM consolidated body samples under each mix proportion is much higher than 0.5 MPa, which meets the technical strength requirements of a construction site, and the microstructure shows that there is an obvious dense and stable block structure inside. The cost of the NUC-FM prepared with an optimized mix proportion is only 34.57 dollars/m^3^, which is far lower than the market purchase price of concrete and cement mortar. It can be predicted that the NUC-FM is an ideal filling grouting material for water-filled karst caves in shield tunnels in water-rich karst areas.

## 1. Introduction

With the rapid development and improvement of China’s transportation infrastructure, an increasing number of subway projects need to be constructed in water-rich karst environments. Yunnan Province, Guizhou Province, and Guangzhou Province in the southwest and southeast of China are the largest karst distribution areas in the world. The karst caves have brought great challenges to engineering construction and urban development. As a complex hydrological and geological system, karst is usually accompanied by the strong development of karst caves. Because of its strong concealment, high complexity, and difficulty in governance, shield tunneling is prone to out-of-control posture, subsidence, water and mud inrushes, and tunnel face instability [1,2,3,4,5], which directly affect the engineering quality and construction safety. At present, typical karst treatment methods include bridge crossing [6,7], pile foundation reinforcement [8,9], excavation backfill [10,11], and grouting filling [12]; however, these karst cave treatment measures have greatly increased the project construction cost. Filling technology is the most widely used karst treatment measure due to its advantages of simple operation, small impact on the surrounding environment, and good governance effect. Among the many factors affecting the grouting effect of karst filling, the grouting material is key.

To date, scholars have carried out much research on karst grouting materials and achieved notable results. For example, taking into account the large amount of karst filling and the need for more filling materials, many researchers [13,14,15,16,17,18,19] have proposed cement-based or cement–clay-based grouting materials, with a wide source of raw materials that are environmentally friendly and at a low price; for more complex water-rich karst grouting, various researchers [20,21,22,23] have proposed chemical grouting materials with better water resistance, faster setting times, and better grouting effects; however, they are relatively more expensive. Generally, there are many kinds of karst filling grouting materials and abundant water-rich karst grouting materials, but unfortunately, there are no reports on the filling grouting materials of water-filled karst caves that meet the requirements of shield tunneling. Moreover, the utilization rate of existing grouting materials for construction waste is low, which is unfavorable to environmental protection and project cost. In contrast to general karst filling grouting materials, the filling materials for water-filled karst caves under shield conditions should not only solve the problems of large slurry consumption for karst cave filling, easy dispersion of materials underwater, and long setting time, but also ensure reasonable strength of the filled solid in order to avoid the problems of cutterhead abrasion or tunnel face instability due to a strength that is too high or too low. In addition, due to various factors in the actual construction process, higher requirements exist for the cost control of filling materials. It is not difficult to see that the existing karst filling grouting materials have difficulty meeting the above requirements at the same time.

In view of this problem, a NUC-FM suitable for grouting and filling water-filled karst caves in metro shield construction is developed. Through laboratory tests, the physical and mechanical properties of the NUC-FM under different proportions are systematically studied, and the microstructure of the slurry consolidation body is studied through SEM and other microscopic analyses. Finally, an economic analysis is carried out, and the mix proportion design is optimized to provide a reference for the further promotion and application of the material and the treatment of karst caves in urban subway construction.

## 2. Mix Proportion Design of NUC-FM

### 2.1. Study on Properties of Undisturbed Shield Mud

The undisturbed shield mud was obtained from a construction site at the Guangzhou Metro, at a sampling depth of 5–10 m. Two kilograms of undisturbed shield mud was dried naturally, ground, and screened through a standard sieve to obtain the particle grading curve of the soil sample, as shown in Figure 1. The basic in situ characteristics of the soil samples are shown in Table 1; the test process meets the relevant requirements of the Chinese standard GB/T 50123-2019 [1] for geotechnical test methods.

A D8advancea25 X-ray diffractometer was used to conduct X-ray diffraction tests on the undisturbed shield mud to determine the main components, as shown in Figure 2. Meanwhile, the microstructure of the undisturbed shield mud was studied by optical microscopy; the observation results are shown in Figure 3.

The XRD test results show that the undisturbed shield mud is mainly composed of quartz, calcite, and illite, whilst also containing a small amount of iron-containing compounds.

The particle grading of the shield mud is good. The observation results of the 50× light microscope show that the larger particles are evenly distributed in the shield mud, which can better serve as the material skeleton, and finer particles fill the gaps between the skeleton particles, which can make the underwater filling material dense and stable. Under a 200× light microscope, there are still large gaps between the larger particles, and the large and small particles are bonded together. The large and small particles are distributed in a state of small groups, indicating that the undisturbed shield mud has a certain viscosity. The above properties of the undisturbed shield mud make it suitable for cement grouting.

### 2.2. Admixtures

(1) Cement

Ordinary Portland cement P. O 42.5 was used in the test; the physical properties and main chemical composition of the cement are listed in Table 2.

(2) Fly ash

Class I fly ash was adopted; the parameters are shown in Table 3.

(3) Flocculating agent: UWB-II anti-washout admixtures of underwater concrete

UWB-II anti-washout admixtures of underwater concrete from the China Petroleum Engineering Technology Research Institute were used in the test, which are mainly composed of sugar polymer compound thickeners, concrete fluidizing agents, and concrete setting time regulators so that the poured concrete mixture is poured in water without segregation, dispersion, or cement loss. The mixture is self-leveling and self-compacting. The physical and mechanical properties and durability of the concrete after setting and hardening are similar to those of ordinary concrete.

(4) Water reducing agent

Polycarboxylic acid superplasticizer was used in the experiment; its solid mass accounts for 40% of the total mass of the solution.

(5) Mineral powder

See Table 4 for the main chemical composition of mineral powder used in the test.

(6) Early strength agent

An early chlorine salt strength agent was selected for the test; the components mainly include calcium chloride, sodium chloride, and aluminum chloride.

### 2.3. Mix Proportion of NUC-FM

For the NUC-FM, it is necessary not only to meet the relevant requirements of grouting filling material for water-filled karst caves, but also to ensure the smooth tunneling of the shield after grouting filling. Therefore, the material should have the characteristics of strong underwater dispersion resistance, good stability, and moderate strength. In this test, according to the performance requirements of underwater construction on underwater cementitious filling materials, a new type of underwater cementitious filling material (NUC-FM) was prepared mainly from waste shield mud, as shown in Table 5.

## 3. Unconfined Compressive Strength Test

### 3.1. Sample Preparation

Setting curing ages of 7 d, 14 d, and 28 d, nine compression samples were made under each mix proportion, and their respective unconfined compressive strengths were measured. NWC-1, NWC-2, NWC-3, NWC-4, and NWC-5 correspond to test Groups G, H, I, J, and K, respectively. When preparing the NUC-FM sample, the shield mud, cement, water, and various admixtures were mixed according to the different mix proportions of each group through a mixer to obtain a paste similar to mixed slurry; then, it was placed into a 70.7 × 70.7 × 70.7 mm mold for vibration and tamping. Finally, the sample was sealed, set in a water tank filled with water, and placed in a standard curing room (20 ± 3 °C, humidity > 95%) for curing to the designated age. The sample preparation process is shown in Figure 4.

The underwater curing process of the NUC-FM has obvious white transparent gel material precipitation, which increases gradually with increasing curing time. After 12 h of underwater curing, the amount of white precipitated gel was basically stable, and an isolation layer was formed on the surface of the NUC-FM.

### 3.2. Test Phenomena

The unconfined compressive strength test was carried out by a microcomputer-controlled electronic universal testing machine. Since there are no specifications for the study of mechanical properties of new underwater aggregate filling materials, the cube compression test referenced the Chinese code DL/T5117-2000 [1] for the underwater testing of non-dispersible concrete and the Chinese standard GB/T50080-2016 [1] for performance test methods of ordinary concrete mixtures. The test method refers to Chinese standard GB/T50081-2019 [1] for test methods of physical and mechanical properties of concrete and Chinese standard JGJ/T70-2009 [1] for test methods of basic properties of building mortar. The test process of selected test blocks is shown in Figure 5.

Figure 5 shows that the failure phenomena and failure modes of the NUC-FM with mixing proportions that are basically the same. There are many vertical cracks on the surface of each test block due to the action of a uniform surface load. When the load is small, the cracks on the surface of the test block are very thin and small, and the cracks are vertically distributed. With the gradual increase in load, the crack width is gradually widened, small cracks are gradually interconnected between cracks (especially at the corner of the test block), and the cracks are distributed in a network. As the load continues to increase, there are many vertical main cracks, and the cracks expand from the surface to the interior. When the load continues to increase, the main cracks penetrate in two parallel planes, and the test block is destroyed.

### 3.3. Test Results and Analysis

The stress–strain relationship under uniaxial compression can effectively reflect the deformation characteristics and failure process of materials at various stress stages [24,25]. The mechanical properties included in the relationship are important parameters for component design and nonlinear analysis. The compressive stress–strain curve (calculated from the load deformation curve) of each group of NUC-FM cube samples obtained from the test is shown in Figure 6 and Figure 7. The unconfined compressive strength of the NUC-FM consolidated body was measured, as shown in Table 6.

Figure 6 shows that the stress–strain relationship of each test block is basically the same, and that the stress–strain of the rising section is similar to a linear change. The slope of the secant between the 1/3 peak stress and the initial stress in the stress–strain curve of the test blocks cured for 28 d and 14 d is basically greater than that of the test block cured for 7 d, indicating that the elastic modulus of the test blocks cured for 28 d and 14 d is improved. With increasing strain, the stress increases and reaches peak stress. The peak stress of the test block with a curing time of 28 d is the largest, followed by that of the test block with a curing time of 14 d, and that of the test block with a curing time of 7 d is the smallest, indicating that the strength of the test block increases with increasing curing time. The peak stress increases of the NWC-1 test blocks after curing for 28 d and 14 d compared with that after curing for 7 d are 6.04 MPa and 4.011 MPa, respectively. The peak stress increases of the NWC-2 test blocks after curing for 28 d and 14 d compared with that after curing for 7 d are 4.02 MPa and 2.762 MPa, respectively. The peak stress increases of the NWC-3 test blocks after curing for 28 d and 14 d compared with that after curing for 7 d are 1.805 MPa and 1.028 MPa, respectively. The peak stress increases of the NWC-4 test blocks after curing for 28 d and 14 d compared with that after curing for 7 d are 3.079 MPa and 0.934 MPa, respectively. The peak stress increases of the NWC-5 test blocks after curing for 28 d and 14 d compared with that after curing for 7 d are 0.728 MPa and 0.281 MPa, respectively. With increasing curing age, each test block has a very obvious late strength development. With increasing strain, the test blocks in each group quickly enter the descending section after reaching peak stress, and the shape of each group is slightly different. The descending curve of the test block cured for 7 days is relatively flat. The stress decreases slowly with increasing strain, and the plastic deformation is larger than that of the other test blocks.

Figure 7 shows that the stress–strain relationship of each test block is basically the same, and the stress–strain of the ascending section is approximately linear. The slope of the curve of the NWC-2 test block is larger than that of the other groups, indicating that the elastic modulus of the NWC-2 test block is the largest, and the ability to resist deformation is the largest. With increasing strain, the stress increases and then reaches peak stress. The peak stress of the NWC-2 group test block is the largest, the peak stress of the NWC-4 group test block is the second largest, and the peak stress of the NWC-5 group test block is the smallest. This shows that the strength of the test block made by the mix proportion of the NWC-2 group is the largest, the mix proportion of the NWC-4 group comes second, and the strength of the test block made by the mix proportion of the NWC-5 group is the smallest; however, the descending section of the NWC-5 group test block is the most gentle, and the deformation ability is better than other mix proportions, indicating that the strength is improved and the corresponding deformation ability is reduced. The above test results show that the strength of the NUC-FM material is greatly affected by the amount of cement under the same shield mud quality. With increasing cement content, the strength gradually increases.

## 4. Microanalysis of NUC-FM Consolidated Body

### 4.1. XRD Analysis

Figure 8 shows the XRD patterns of the undisturbed shield mud and NUC-FM solidified body with a curing age of 28 d. The mineral composition of undisturbed shield mud mainly includes quartz (SiO_2_), calcite (CaCO_3_), and illite (KAl_2_[(SiAl)_4_O_10_]·(OH)_2_·nH_2_O), with a small amount of hematite (Fe_2_O_3_). Compared with the undisturbed shield mud, the peak shape of the NUC-FM consolidation sample is basically unchanged, but there are humps and weak diffraction peaks of C-S-H gel in the range of 2*θ* = 29°~33°, and the peak height of the maximum peak value of calcite (approximately 29°) increases, indicating that the NUC-FM consolidation has generated calcium silicate hydrate gel and calcite crystal. Calcium silicate hydrate is the main product of the hydration reaction of raw materials such as cement and mineral powder [26]. Reference [27] noted that after mixing soil with cement, cement hydration to generate C-S-H was the main contributor to the strength of solidified soil. As the hydration product C-S-H gel is amorphous, there is no obvious characteristic peak, but the corresponding dispersion peak can be observed in the range of 2*θ* = 29°~33°. According to Zhang et al., the broad peak formed by the gel will cover up other crystal peaks, so the peak strength related to quartz and illite can be observed to decrease. At the same time, some minerals in the shield mud participate in the volcanic ash reaction, which is also one of the reasons for the decrease in the characteristic peak strength of quartz and illite.

### 4.2. Analysis of Microstructure Characteristics

Figure 9a,b shows SEM images of undisturbed shield mud and the Group G consolidated body test block enlarged by 500 times, respectively. The undisturbed shield mud is mainly a layered structure, and there are irregular massive soil particles. The particles are arranged in a disordered order and have large dispersion, and there are many overhead pores. There are no other substances in the overhead pores. After the undisturbed shield mud is stabilized by cement, flocculant, fly ash, and other admixtures, flocculant, anchor flake and fibrous gel products appear on the surface of the consolidation body. The soil particles are closely bonded, and the particle gap is effectively filled. The total volume of pores is greatly reduced. The material structure is more compact in the SEM image, and the macroscopic performance shows that the NUC-FM consolidation body has a certain strength.

The influencing factors of the NUC-FM strength include the soil particle size distribution, undisturbed shield mud void ratio, types of curing agents, sample preparation methods (stirring conditions, compaction conditions), and curing conditions (temperature, humidity, age, etc.). The fundamental reason for the difference in strength is the difference in the microstructure characteristics of the consolidation body. To further explore the micromorphology of the new underwater coagulation filling material, the consolidation samples of each test group are magnified 10,000 times, as shown in Figure 10.

Figure 10 shows that there is a large number of fibrous materials in the SEM images of each experimental group that connect soil particles and particle clusters to form the interactive spatial structure of soil skeleton fibers. The formation of NUC-FM strength is mainly due to the filling effect and cementation, which is reflected in the following two aspects. First, due to the large amount of CaO in raw materials (such as cement and mineral powder), calcium oxide reacts with water to generate Ca(OH)_2_; however, there is no Ca(OH)_2_ diffraction peak in the XRD pattern of the NUC-FM, indicating that Ca(OH)_2_ participates in the secondary hydration reaction and provides an alkaline environment. A series of cement hydration products (such as hydrated calcium silicate) are produced. The reaction equation is as follows: $x\mathrm{Ca}{\left(\mathrm{OH}\right)}_{2}+{\mathrm{SiO}}_{2}+\left(n-1\right)H_{2}O=x\mathrm{CaO}\cdot {\mathrm{SiO}}_{2}\cdot nH_{2}O$, which fills the pores of NUC-FM, making the structure denser. Second, the generated hydration products not only have a filling effect, but also have high cementation strength and can connect the shield mud particles with each other. The cementation makes NUC-FM form a whole with high compactness, thereby increasing its strength. Figure 10b shows clearly visible acicular and prismatic calcium carbonate crystals embedded between cemented shield mud particles. According to its morphology and XRD analysis results, aragonite calcium carbonate crystals are formed during the formation of a consolidated body of new underwater aggregate filling material. Because the aragonite crystal phase is a metastable crystal phase of calcium carbonate, it easily changes to calcite calcium carbonate [28]. Therefore, with increasing age, the aragonite calcium carbonate crystal in the sample changes to calcite calcium carbonate crystals, and the needle and columnar crystal forms of aragonite are still retained. Therefore, only the diffraction peak of calcite appears in the XRD pattern.

Compared with other test groups, the SEM images of the consolidated body samples in Group H show that the large pores were significantly reduced, the ultra-large pores basically disappeared, and there were only a small number of small pores. From the perspective of cement solidification, when the cement content is low, there are fewer cementation products, so there is not much effective bonding between shield mud particles. With increasing cement content, the cementation products increase obviously, and more effective bonding is formed between shield mud particles. Since the cracks between matrix aggregate interfaces are the main cause of concrete damage, the strength decreases due to the existence of a large number of cracks and pores [29,30]. With increasing cement content, the cracks and pores of the NUC-FM microstructure decrease. The higher degree of polymerization gelation makes the microdensity larger, more continuous, and more compact. The external compressive strength is increased by blocking the path of cracks. Therefore, the strength of Group H is higher than that of the other test groups. This phenomenon is consistent with the above unconfined compressive strength test results.

## 5. Mix Proportion Optimization of NUC-FM

Since the technical requirement of the NUC-FM is to ensure the smooth progress of the shield machine under the premise of ensuring that the underwater filling is not dispersed, the strength requirement after filling is greater than or equal to the strength of the undisturbed soil; that is, the strength after filling is approximately 0.5 MPa. From the above mix design of the NUC-FM and the corresponding mechanical properties test, it can be seen that the minimum 7 d average strength of the NUC-FM prepared according to the mix proportion given in Table 6 is 1.734 MPa, which is much higher than the requirement of 0.5 MPa. At the same time, considering that many kinds of admixtures are needed in NUC-FM configuration, excessive strength will cause considerable waste. Therefore, based on a comprehensive consideration of the strength requirements and preparation cost of the NUC-FM, the mixing proportion of the NUC-FM was optimized, a test block was made, and a 7 d strength test was carried out. The failure mode of the test block is shown in Figure 10, and the optimized combination and test results are shown in Table 7.

Figure 11 shows that the failure phenomenon of the NUC-FM test block after optimization of the mix proportion is obvious in the 7 d compressive strength test. The failure mode is shown as a number of obliquely penetrated fracture failure surfaces, and one or two sides of the test block are spalled overall. After adjusting the mix proportion, the average pressure resistance of the test block can reach 0.59 MPa, which meets the technical strength requirements of the construction site.

On the basis of the above experiments, an economic evaluation of NUC-FM after optimization of the mix proportion was carried out. The raw material prices are detailed in Table 8.

The cost of the NUC-FM is estimated. Preparation of 1 m^3^ NUC-FM is approximately 34.57 dollars. At the same time, after investigation, the price of C20 commercial concrete is 94.29–110.01 dollars/m^3^, and the price of cement mortar is 55.00–70.72 dollars/m^3^. The preparation cost of the NUC-FM is far lower than the market price of concrete and cement mortar, which offers good engineering application prospects.

## 6. Conclusions

In this paper, an inexpensive and environmentally friendly grouting material was developed to overcome the shortcomings of traditional grouting materials for the treatment of water-filled karst caves in subway and urban comprehensive gallery construction [31]. The mix proportion was designed, and a series of tests were conducted to check the mechanical property parameters of the material and analyze its microstructure characteristics. The major conclusions are as follows:

(1) The particle gradation of the original shield mud is relatively reasonable. Under an optical microscope, it can be seen that the particle gradation of the shield mud is relatively uniform, the particles with large particle size are evenly distributed inside the shield mud (which can better play a skeleton role), and there are ultrafine particles to fill the gap between large-sized particles. Based on this shield mud, the prepared NUC-FM can be dense and stable.

(2) The strength of the NUC-FM material is greatly affected by the amount of cement under the same shield mud quality conditions, and the strength gradually increases with increasing cement content. Meanwhile, with increasing curing age, the NUC-FM material has a very obvious late strength development; the strength of the NUC-FM consolidation test block under each mix proportion is much higher than 0.5 MPa, which meets the strength technical requirements of the construction site.

(3) The excellent synergistic interactions between shield mud, cement, flocculant, fly ash, and other raw materials can lead to a grouting filling material of NUC-FM with superior performance. Through XRD test results and SEM images, it can be seen that SiO_2_ in raw materials such as original shield mud, cement, fly ash, and mineral powder reacts with Ca(OH)_2_ to generate calcium silicate hydrate. On the one hand, calcium silicate hydrate will be filled in each pore of shield mud to reduce the porosity of the soil. On the other hand, calcium silicate hydrate has a cementation effect, cementing shield mud particles together and forming a combined network structure with them. These effects make the NUC-FM structure more stable and form a certain strength, which makes it suitable for filling water-filled caves.

(4) The laboratory mix proportion test shows that the NUC-FM is a green, environmentally friendly, and inexpensive grouting material that can meet the requirements of water-rich karst filling grouting and subsequent shield tunneling. Under the optimal mix proportion, the material has the characteristics of non-dispersion under water and moderate strength of the consolidation body. The optimized mix proportion is selected to prepare 1 m^3^ new underwater coagulation filling material, and the cost is approximately USD 34.57, which is far lower than the market purchase price of concrete and cement mortar; it also has superior properties as compared with traditional grouting materials in terms of the grouting filling engineering of water-filled karst caves. Considering the increasing number of subway construction projects, an increasing number of shield tunnels in water-rich karst areas will be constructed. Thus, it can be predicted that NUC-FM has broad application prospects.

In contrast to previous studies, NUC-FM takes shield mud as the base material, realizes the grouting and filling of water-filled karst cave by optimizing admixture and mix proportion, and ensures the safety of tunnel construction. At the same time, the secondary utilization of shield mud reduces environmental pollution and construction cost.

## Figures and Tables

**Figure 1 materials-15-02938-f001:**
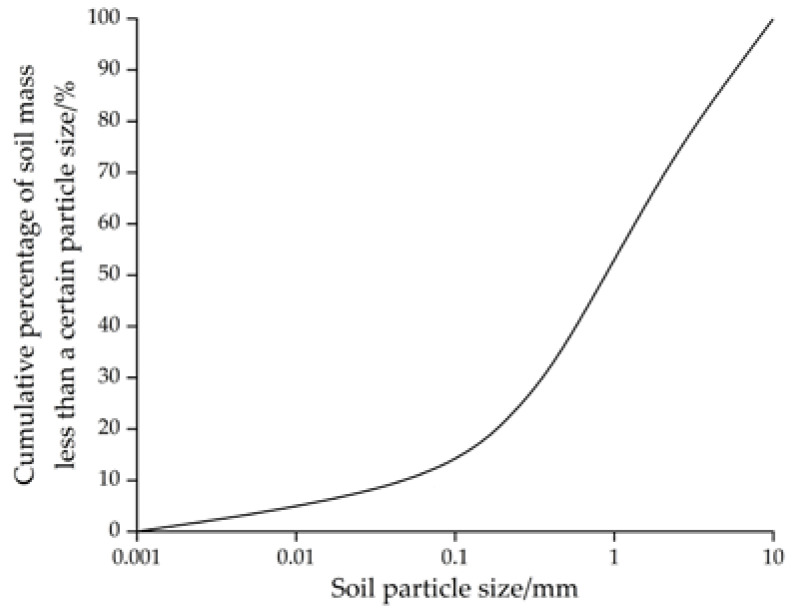
Particle size distribution curve of shield mud.

**Figure 2 materials-15-02938-f002:**
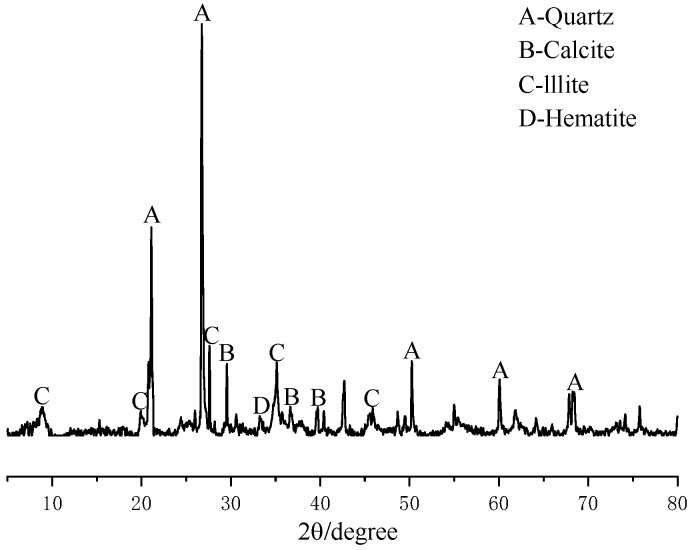
XRD diffractograms of shield mud.

**Figure 3 materials-15-02938-f003:**
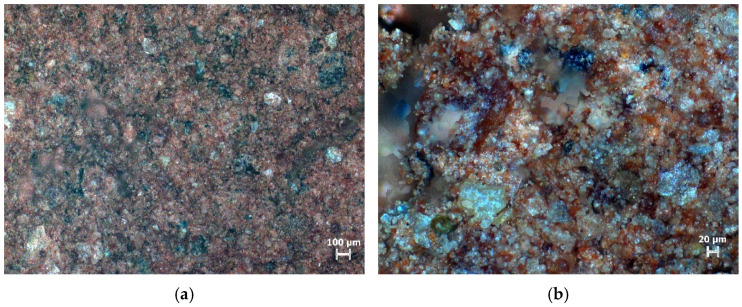
Optical microscope observation results. (**a**) 50 times magnification; (**b**) 200 times magnification.

**Figure 4 materials-15-02938-f004:**
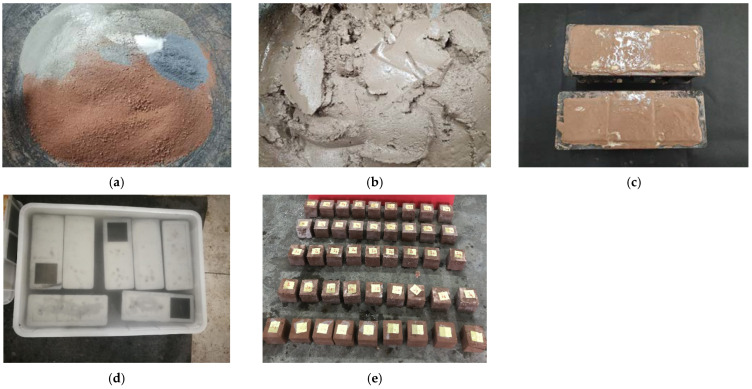
Test sample preparation process. (**a**) Preparing raw materials; (**b**) Mixing with water; (**c**) Puting in the mold; (**d**) Water curing; (**e**) NUC-FM samples.

**Figure 5 materials-15-02938-f005:**
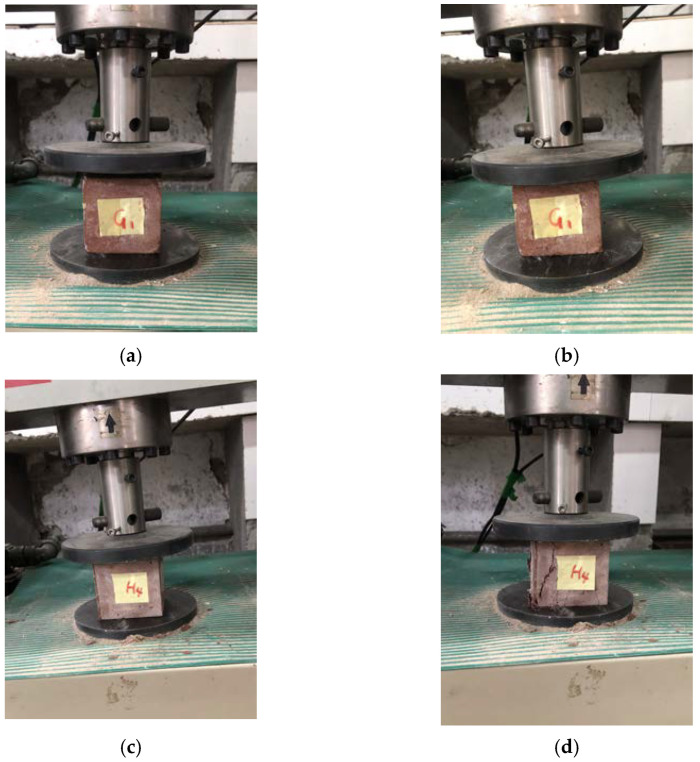
Unconfined compressive strength test of NUC-FM. (**a**) Compressive strength test of Group G test block; (**b**) Group G test block destruction; (**c**) Compressive strength test of Group H test block; (**d**) Group H test block destruction.

**Figure 6 materials-15-02938-f006:**
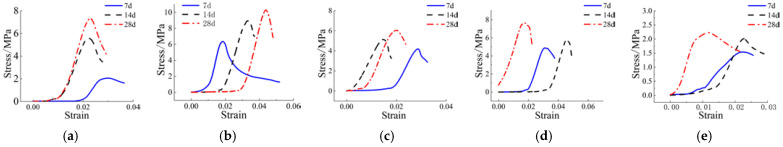
Stress–strain curves of test blocks at different ages. (**a**) Group G; (**b**) Group H; (**c**) Group I; (**d**) Group J; (**e**) Group K.

**Figure 7 materials-15-02938-f007:**
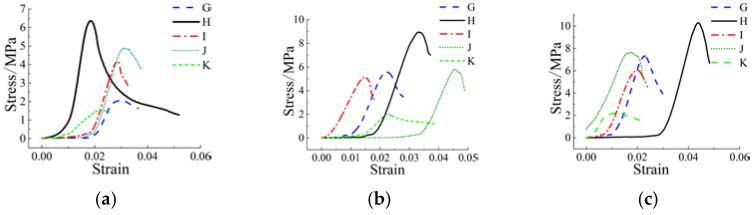
Stress–strain curves at the same age. (**a**) 7 d; (**b**) 14 d; (**c**) 28 d.

**Figure 8 materials-15-02938-f008:**
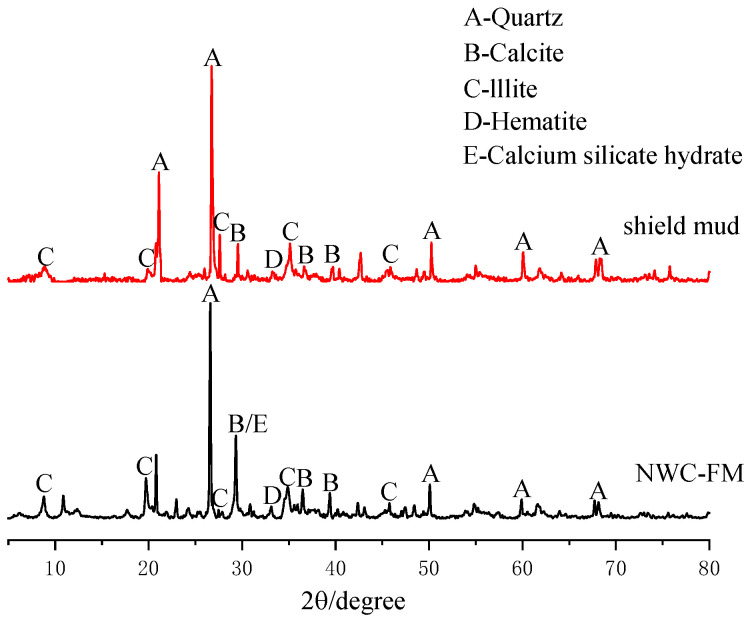
XRD diagram of undisturbed shield mud and consolidated body of NUC-FM.

**Figure 9 materials-15-02938-f009:**
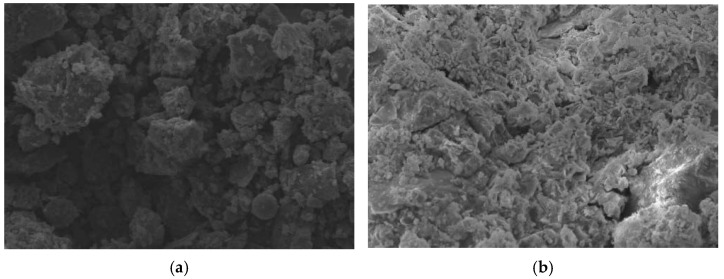
SEM images (500 times). (**a**) Undisturbed shield mud; (**b**) Group G test block.

**Figure 10 materials-15-02938-f010:**
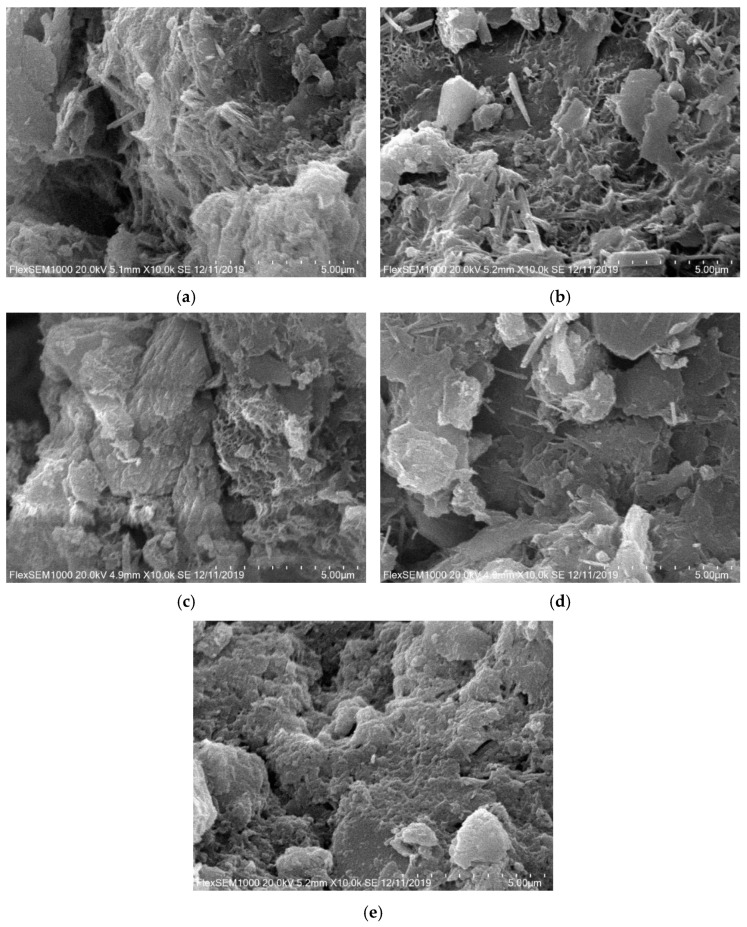
SEM images of consolidated body of NUC-FM (10,000 times). (**a**) Group G; (**b**) Group H; (**c**) Group I; (**d**) Group J; (**e**) Group K.

**Figure 11 materials-15-02938-f011:**
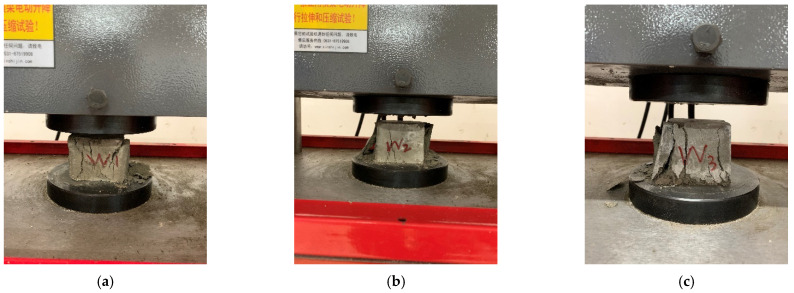
Sample destruction. (**a**) W_1_ test block destruction. (**b**) W_2_ test block destruction. (**c**) W_3_ test block destruction.

**Table 1 materials-15-02938-t001:** In situ properties of shield mud.

Moisture Content (%)	Liquid Limit (%)	Plastic Limit (%)	Void Ratio	Density (g·cm^−3^)
18.81	43.1	24.9	1.1	1.9

**Table 2 materials-15-02938-t002:** Physical properties and main chemical composition of cement.

Ignition Loss (%)	Specific Surface Area (m^2^/kg)	Initial Setting Time (min)	Final Setting Time (min)	Main Chemical Composition (%)				
				CaO	SiO_2_	Al_2_O_3_	Fe_2_O_3_	MgO
2.12	350	235	319	40.41	19.79	7.67	2.66	2.06

**Table 3 materials-15-02938-t003:** Physical properties and main chemical composition of fly ash.

Ignition Loss (%)	Fineness (%)	Specific Surface Area (m^2^/kg)	28 d Activity Index (%)	Density (g·cm^3^)	Main Chemical Composition (%)				
					Fe_2_O_3_	CaO	MgO	Al_2_O_3_	SiO_2_
**2.86**	**12**	**438**	**53**	1.12	3.87	2.27	0.81	29.09	53.36

**Table 4 materials-15-02938-t004:** Main chemical composition of mineral powder (%).

SiO_2_	Fe_2_O_3_	Al_2_O_3_	CaO	MgO	TiO_2_	K_2_O	Na_2_O
33.15	0.26	15.6	36.78	10.21	0.59	0.47	0.37

**Table 5 materials-15-02938-t005:** Mix Proportion of NUC-FM (kg).

Component	Number				
	NWC-1	NWC-2	NWC-3	NWC-4	NWC-5
Water	0.429	0.357	0.302	0.272	0.373
Cement	0.286	0.357	0.175	0.227	0.162
Fly ash	0.057	0.071	0.038	0.045	0.032
Mineral powder	0.021	0.021	0.021	0.018	0.013
Flocculating agent	0.014	0.014	0.032	0.037	0.024
Shield mud	1.000	1.000	1.000	1.000	1.000
Water reducing agent	0.010	0.010	0.008	0.007	0.008
Early strength agent	0.010	0.010	0.008	0.007	0.008

**Table 6 materials-15-02938-t006:** Compressive strength of NUC-FM cube specimen.

Curing Age/d	Number	Average Compressive Strength/MPa
7 d	NWC-1	2.112
14 d		6.123
28 d		8.152
7 d	NWC-2	6.248
14 d		9.010
28 d		10.268
7 d	NWC-3	4.320
14 d		5.348
28 d		6.125
7 d	NWC-4	5.155
14 d		6.089
28 d		8.234
7 d	NWC-5	1.734
14 d		2.015
28 d		2.462

**Table 7 materials-15-02938-t007:** Optimized mix proportion and 7 d compressive strength test results.

**Optimized Mix Proportion NWC-W (kg)**	**Shield Mud**	**Water**	**Cement**	**Fly Ash**
	**1.0**	**0.351**	**0.141**	**0.041**
Strength (MPa)	**NWC-w1**	**NMC-w2**	**NWC-w3**	**Mean Value**
	0.61	0.56	0.59	0.59

**Table 8 materials-15-02938-t008:** NUC-FM Raw Material Price Details.

Materials	Water	Cement	Fly Ash	Mineral Powder	Flocculating Agent	Water Reducing Agent
Unit price (dollar/t)	90.36	32.69	75.43	1414.40	562.62	612.91

## Data Availability

Not applicable.

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
