# Peer review of "Experimental Study on the Mix Proportion and Mechanical Properties of New Underwater Cementitious Filling Materials"

_materials, 2022, doi:10.3390/ma15082938_

Round 1

Reviewer 1 Report

The study "Experimental Study on the Mix Proportion and Mechanical Properties of New Underwater Cementitious Filling Materials" is well written.

Few comments are give below to improve the quality of the article.

  1. There is need to insert quantitivate results in the abstract.
  2. Introduction section need improvement to further highlight the importance of this study.
  3. Figure 6 shows that the stress–strain relationship of each test block is basically the same, and the stress–strain of the rising section is similar to a linear change. --- There is need to give approporate reasons for this similar behavior.
  4. The descending curve of the test block cured for 7 days is relatively flat. The stress decreases slowly with increasing strain, and the plastic deformation 201 is larger than that of the other test blocks. --- Further no reasons are provided for this phenomenon.
  5. Its appropriate to compare results with the previous studies.
  6. Conclusions are too long, its better to write to the point.

Author Response

Response to reviewers

Dear reviewers:
We are very grateful to the reviewers for your comments on the manuscript, which enables us to further improve the quality and format of the paper. Each suggested revision and comment, brought forward by the reviewers were carefully considered and corrected.
The following is a response to the revised content of the manuscript based on the comments of the reviewers . The font with a green background indicates the revision of the manuscript based on the comments of the reviewers. The gray background font indicates that the content in the original manuscript is deleted. The yellow background font indicates the content added in the original manuscript. Should you have any questions, please contact us without hesitate.
Thanks again to all reviewers for your hard work and patience.

1.Comments: There is need to insert quantitivate results in the abstract.
1.Response: According to your suggestion, we have added quantitative analysis to the summary. The revised summary is as follows:
Considering that it is difficult for traditional materials to meet the filling grouting requirements of water-filled karst caves and subsequent shield tunneling at the same time, an environmentally friendly and controllable new underwater cementitious filling material (NWC-FM) is developed with abandoned shield mud as the basic raw material. Through laboratory tests, the mechanical property parameters of NWC-FM are tested, and its micromechanism is analyzed. The research results show that there is excellent synergistic interactions among shield mud, cement, flocculant, fly ash and other raw materials. The NWC-FM grouting filling material with superior performance can be prepared when the water binder ratio is between 0.45 ~ 0.6 and the water consumption is between 270 ~ 310kg / m3. It has the characteristics of nondispersion under water and moderate consolidated body strength. The compressive strength of the NWC-FM consolidated body samples under each mix proportion is much higher than 0.5 MPa, which meets the technical strength requirements of a construction site, and the microstructure shows that there is an obvious dense and stable block structure inside. The cost of the NWC-FM prepared with an optimized mix proportion is only 34.57dollar/m3 ,which is far lower than the market purchase price of concrete and cement mortar. It can be predicted that the NWC-FM is an ideal filling grouting material for water-filled karst caves in shield tunnels in water-rich karst areas.

2.Comments: Introduction section need improvement to further highlight the importance of this study.
2.Response: With the rapid development and improvement of China’s transportation infrastructure, an increasing number of subway projects need to be constructed in water-rich karst environments. Yunnan Province, Guizhou Province and Guangzhou Province in the southwest and southeast of China are the largest karst distribution areas in the world. The karst caves have brought great challenges to engineering construction and urban development. As a complex hydrological and geological system, karst is usually accompanied by the strong development of karst caves. Because of its strong concealment, high complexity and difficulty in governance, shield tunneling is prone to out of control posture, subsidence, water and mud inrushes and tunnel face instability [1-5], which directly affect the engineering quality and construction safety. At present, typical karst treatment methods include bridge crossing [6,7], pile foundation reinforcement [8,9], excavation backfill [10-11] and grouting filling [12], these karst cave treatment measures have greatly increased the project construction cost. In karst cave treatment measures, filling technology is the most widely used in many karst governance methods due to its advantages of simple operation, small impact on the surrounding environment and good governance effect. Among the many factors affecting the grouting effect of karst filling, the grouting material is one of the key factors.
To date, scholars have carried out much research on karst grouting materials and achieved notable results. For example, taking into account the large amount of karst filling and the need for more filling materials, Ni et al. [13-19] proposed cement-based or cement–clay-based grouting materials with a wide source of raw materials that were environmentally friendly at a low price; for more complex water-rich karst grouting, Li et al. [20-23] proposed chemical grouting materials with better water resistance, faster setting times and better grouting effects, but they were relatively more expensive. Generally, there are many kinds of karst filling grouting materials and abundant water-rich karst grouting materials, but unfortunately, there are no reports on the filling grouting materials of water-filled karst caves that meet the requirements of shield tunneling. Moreover, the utilization rate of existing grouting materials for construction waste is low, which is unfavorable to environmental protection and project cost. Different from general karst filling grouting materials, the filling materials for water-filled karst caves under shield conditions should not only solve the problems of large slurry consumption for karst cave filling, easy dispersion of materials underwater and long setting time but also ensure the reasonable strength of the filled solid to avoid the problems of cutterhead abrasion or tunnel face instability due to a strength that is too high or too low. In addition, due to various factors in the actual construction process, higher requirements exist for the cost control of filling materials. It is not difficult to see that the existing karst filling grouting materials have difficulty meeting the above requirements at the same time.
In view of this problem, a new underwater cementitious filling material (NWC-FM) suitable for grouting and filling water-filled karst caves in metro shield construction is developed. Through laboratory tests, the physical and mechanical properties of the new underwater cementitious filling material under different proportions are systematically studied, and the microstructure of the slurry consolidation body is studied through SEM and other microscopic analyses. Finally, an economic analysis is carried out, and the mix proportion design is optimized to provide a reference for the further promotion and application of the material and the treatment of karst caves in urban subway construction.

3.Comments: Figure 6 shows that the stress–strain relationship of each test block is basically the same, and the stress–strain of the rising section is similar to a linear change. --- There is need to give appropriate reasons for this similar behavior.
3.Response: In this experiment, the fine aggregate of traditional concrete is replaced by shield mud, and the grouting filling of water filled karst cave is realized through the selection of admixture. Its basic material is similar to that of traditional cement concrete, so its mechanical properties are roughly the same as that of traditional cement concrete in the stress-strain curve, and the linear change in the rising section.

4.Comments: The descending curve of the test block cured for 7 days is relatively flat. The stress decreases slowly with increasing strain, and the plastic deformation 201 is larger than that of the other test blocks. --- Further no reasons are provided for this phenomenon.
4.Response: The strength of the test block is low after curing for 7 days. In order to prevent the damage of the test block, when the stress-strain curve of the test block is obtained by the universal testing machine, the loading and unloading force is small, and the micro plasticity of the test block is also reflected in the stress-strain curve, so the decline curve of the test block is more flat. It can be seen from the test ratio that the relative content of cementitious materials (cement, fly ash and mineral powder) in nwc-2 is high. The research conclusions of Joo ha Lee, Achintyamugdha Sharma and others prove that cementitious materials improve the plasticity of materials, so their plastic deformation is relatively large[1-2].
[1] Lee J H , Yoon Y S . The effects of cementitious materials on the mechanical and durability performance of high-strength concrete[J]. KSCE Journal of Civil Engineering, 2014.
[2] Sharma A , Sirotiak T , Stone M L , et al. Effects of Cement Changes and Aggregate System on Mechanical Properties of Concrete 2. 2020.

5.Comments: Its appropriate to compare results with the previous studies.
5.Response: According to your suggestion, we have increased the comparison between the research conclusions and previous studies to improve the persuasiveness of the research.
Compared with previous studies, "NWC-FM" takes shield mud as the base material, realizes the grouting and filling of water filled karst cave by optimizing admixture and mix proportion, and ensures the safety of tunnel construction. At the same time, the secondary utilization of shield mud reduces environmental pollution and construction cost.

6.Comments: Conclusions are too long, its better to write to the point.
6.Response: Based on your comments, I have further generalized and summarized the conclusions of the manuscript.
(1) The particle gradation of the original shield mud is relatively reasonable. Under an optical microscope, it can be seen that the particle gradation of the shield mud is relatively uniform, the particles with large particle size are evenly distributed inside the shield mud, which can better play a skeleton role, and there are ultrafine particles to fill the gap between large particle size particles. Based on this shield mud, the prepared NWC-FM can be dense and stable.
(2) The strength of the NWC-FM material is greatly affected by the amount of cement under the same shield mud quality conditions, and the strength gradually increases with increasing cement content. At the same time, with increasing curing age, the NWC-FM material has a very obvious late strength development; the strength of the NWC-FM consolidation test block under each mix proportion is much higher than 0.5 MPa, which meets the strength technical requirements of the construction site.
(3) The excellent synergistic interactions among shield mud, cement, flocculant, fly ash and other raw materials can lead to a grouting filling material of NWC-FM with superior performance. Through XRD test results and SEM images, it can be seen that SiO2 in raw materials such as original shield mud, cement, fly ash and mineral powder reacts with Ca(OH)2 to generate calcium silicate hydrate. On the one hand, calcium silicate hydrate will be filled in each pore of shield mud to reduce the porosity of the soil. On the other hand, calcium silicate hydrate has a cementation effect, cementing shield mud particles together and forming a combined network structure with them. These effects make the NWC-FM structure more stable and form a certain strength, which makes it suitable for filling water-filled caves.
(4) The laboratory mix proportion test shows that the NWC-FM is a green, environmentally friendly and inexpensive grouting material that can meet the requirements of water-rich karst filling grouting and subsequent shield tunneling. under the optimal mix proportion, the material has the characteristics of no dispersion under water and moderate strength of the consolidation body. At the same time, the optimized mix proportion is selected to prepare 1 m3 new underwater coagulation filling material, and the cost is approximately CNY 220, which is far lower than the market purchase price of concrete and cement mortar and has superior properties as compared with traditional grouting materials in terms of the grouting filling engineering of water-filled karst caves. Considering the increasing number of subway construction projects, an increasing number of shield tunnels in water-rich karst areas will be constructed. It can be predicted that NWC-FM has broad application prospects.

Reviewer 2 Report

The article presents a reasearch on a conglomerate with cementitious binder to be used as a filling for karst caves full of water. The analytical process aimed at evaluating the optimal mix for the intended function is well described and scientifically valid.

It seems to me that the language and terminology – especially in some paragraphs – need to be improved.

Some of the references are articles written in Chinese and their title is cited in English, without the specification of the original title or language of the article. This make it difficult to find the references, also because their DOI is missing.

Since an acronym has been defined, it should always substitute the full name «new underwater cementitious filling material» after its first appearance in the article.

The authors can find some minor comments and misspelling in the attached file.

Author Response

Response to reviewers

Dear reviewers:
We are very grateful to the reviewers for your comments on the manuscript, which enables us to further improve the quality and format of the paper. Each suggested revision and comment, brought forward by the reviewers were carefully considered and corrected.
The following is a response to the revised content of the manuscript based on the comments of the reviewers . The font with a green background indicates the revision of the manuscript based on the comments of the reviewers. The gray background font indicates that the content in the original manuscript is deleted. The yellow background font indicates the content added in the original manuscript. Should you have any questions, please contact us without hesitate.
Thanks again to all reviewers for your hard work and patience.

1.Comments: 12 The acronym NWC-FM doesn’t match the name «new underwater cementitious filling material», it should be NUC-FM or NUWC-FM.
1.Response: According to your suggestion, we have revised " NWC-FM " in the text to "NUC-FM".

2.Comments:
32 It’s not clear what «head planting» is. Please, specify the meaning or correct/improve the translation.
2.Response: "Head planing" means that the head of the shield machine deviates from the design axis and settles downward. According to your suggestion, I will change "head planning" to "out of control posture".
Because of its strong concealment, high complexity and difficulty in governance, shield tunneling is prone to out of control posture, subsidence, water and mud inrushes and tunnel face instability [1-5], which directly affect the engineering quality and construction safety.

3.Comments:
The authors wrote «Ni et al. [13-19]». This kind of reference style is inappropriate, since Ni is the first author of the first reference, but not of the other. Please correct with another reference style (e.g. “various authors” or “various researchers”, …)
3.Response: Many researchers [13-19] proposed cement-based or cement–clay-based grouting materials with a wide source of raw materials that were environmentally friendly at a low price; for more complex water-rich karst grouting

4.Comments:
45-46 The authors wrote «Li et al. [20-23]». Again, the reference style is inappropriate, see the previous comment.
4.Response: various researchers [20-23] proposed chemical grouting materials with better water resistance.

5.Comments:
59 As in the abstract, the acronym NWC-FM doesn’t match the name «new underwater cementitious filling material».
5.Response: According to your suggestion, the full text was checked and "NWC-FM" was revised to "NUC-FM".

6.Comments:
61-62 Since an acronym has been defined, it should always substitute the full name «new underwater cementitious filling material» here and in the rest of the article.
6.Response: According to your suggestion, we have abbreviated the words except "new underwater cemented filling material" in the definition.

7.Comments:
117 The number of atoms in the molecules should be indicated in the subscript.
7.Response: According to your suggestion, we have corrected the number of atoms in the molecular number.

8.Comments:
189-195 There are capitalized words following semicolons. Please correct the semicolons in full stops or remove the capitalization of the words.
8.Response: According to your suggestion, we checked the punctuation mark and word case, and corrected the errors.
The peak stress increases of the NWC-1 test block after curing for 28 d and 14 d compared with that after curing for 7 d are 6.04 MPa and 4.011 MPa, respectively. The peak stress increases of the NWC-2 test block after curing for 28 d and 14 d compared with that after curing for 7 d are 4.02 MPa and 2.762 MPa, respectively. The peak stress increases of the NWC-3 test block after curing for 28 d and 14 d compared with that after curing for 7 d are 1.805 MPa and 1.028 MPa, respectively. The peak stress increases of the NWC-4 test block after curing for 28 d and 14 d compared with that after curing for 7 d are 3.079 MPa and 0.934 MPa, respectively. The peak stress increases of the NWC-5 test block after curing for 28 d and 14 d are 0.728 MPa and 0.281 MPa, respectively compared with that after curing for 7 d. With increasing curing age, each test block has a very obvious late strength development. With increasing strain, the test blocks in each group quickly enter the descending section after reaching the peak stress, and the shape of each group is slightly different. The descending curve of the test block cured for 7 days is relatively flat. The stress decreases slowly with increasing strain, and the plastic deformation is larger than that of the other test blocks.

Reviewer 3 Report

Dear Authors,

This article including experimental study on the mix proportion and mechanical properties of new underwater cementitious filling materials. As a result, an inexpensive and environmentally friendly grouting material was developed to overcome the shortcomings of traditional grouting materials for the treatment of water-filled karst caves in subway shield construction. Considering the increasing number of subway construction projects, publication and results are of significant economic and scientific importance.

So, from my point of view,  this is an interesting article and has a very high scientific value. Compared to other published materials, the topic is original.

In my opinion, the article requires minor substantive corrections, as well as the layout and additions.

After supplements, I recommend this article for publication in Materials.

Below are my comments:

Why was such a small sample of 2 kg taken for testing? It was possible to collect more material from different places and then, according to the rules of sample reduction, average it to a smaller one.

Figure 1. Particle size distribution curve of shield mud should be corrected. The beginning of both axes of the coordinate system starts at 0. Please correct the units.

Please pay attention to all tables and figures to correct /units as required by the editor. Please pay attention to superscripts and subscripts.

In table 1, values are given in tenths, hundredths and thousandths of one. It is enough to enter all of them in tenths. Please note the other tables.

Table 1. Why the liquid limit and plastic limit are given as a percentage, since the unit is Pa? Please explain the method of measurement, because most readers do not have access to Chinese standards.

Please format the text correctly. Keep the spaces. Do not separate table 5 by pages.

In lines 110 and 111 it says: „The physical and mechanical properties and durability of the concrete after setting and hardening are similar to those of ordinary concrete”.  Is it simple concrete or cement mix? With what aggregate, up to 10 mm, since the grain size of the feed is up to 10 mm. Table 6 shows that average compressive strength max = 10 MPa. For simple concretes, this strength is about 30-50 MPa. Please justify the difference in strength.

In table 8 and in the text, costs are expressed in yuan/t. It will be easier to compare the costs in USD.

Kind regards.

Author Response

Response to reviewers

Dear reviewers:
We are very grateful to the reviewers for your comments on the manuscript, which enables us to further improve the quality and format of the paper. Each suggested revision and comment, brought forward by the reviewers were carefully considered and corrected.
The following is a response to the revised content of the manuscript based on the comments of the reviewers . The font with a green background indicates the revision of the manuscript based on the comments of the reviewers. The gray background font indicates that the content in the original manuscript is deleted. The yellow background font indicates the content added in the original manuscript. Should you have any questions, please contact us without hesitate.
Thanks again to all reviewers for your hard work and patience.

General Comments:

Dear Authors,
This article including experimental study on the mix proportion and mechanical properties of new underwater cementitious filling materials. As a result, an inexpensive and environmentally friendly grouting material was developed to overcome the shortcomings of traditional grouting materials for the treatment of water-filled karst caves in subway shield construction. Considering the increasing number of subway construction projects, publication and results are of significant economic and scientific importance.
So, from my point of view, this is an interesting article and has a very high scientific value. Compared to other published materials, the topic is original.
In my opinion, the article requires minor substantive corrections, as well as the layout and additions.
After supplements, I recommend this article for publication in Materials.
Comments: Thank you for your valuable suggestions, which are of great help to improving the rigor and scientific nature of the manuscript. The following is our response to your comments.

1.Comments: Why was such a small sample of 2 kg taken for testing? It was possible to collect more material from different places and then, according to the rules of sample reduction, average it to a smaller one.
1.Response: In the actual test process, several soils were excavated for detection, but only 2kg was selected for description in this paper. Figure 1 shows the actual soil borrowing and drying photos.

2.Comments:
Figure 1. Particle size distribution curve of shield mud should be corrected. The beginning of both axes of the coordinate system starts at 0. Please correct the units.
2.Response: According to your suggestion, we have modified the particle grading curve of shield mud, as shown in Figure 2.

3.Comments:
Please pay attention to all tables and figures to correct /units as required by the editor. Please pay attention to superscripts and subscripts.
3.Response:According to your suggestion, we have revised the format of tables and numbers.

4.Comments:
In table 1, values are given in tenths, hundredths and thousandths of one. It is enough to enter all of them in tenths. Please note the other tables.
4.Response: According to your suggestion, all the figures in Table 1 have been accurate to one tenth.

5.Comments:
Table 1. Why the liquid limit and plastic limit are given as a percentage, since the unit is Pa? Please explain the method of measurement, because most readers do not have access to Chinese standards.
5.Response: Plastic limit is the lower limit moisture content of fine-grained soil in plastic state, and liquid limit is the upper limit moisture content of fine-grained soil in plastic state, both of which are expressed in percentage.

6.Comments:
Please format the text correctly. Keep the spaces. Do not separate table 5 by pages.
6.Response: According to your suggestion, table 5 has been placed on one page.

7.Comments:
In lines 110 and 111 it says: „The physical and mechanical properties and durability of the concrete after setting and hardening are similar to those of ordinary concrete”. Is it simple concrete or cement mix? With what aggregate, up to 10 mm, since the grain size of the feed is up to 10 mm. Table 6 shows that average compressive strength max = 10 MPa. For simple concretes, this strength is about 30-50 MPa. Please justify the difference in strength.
7.Response: The above description compares the physical and mechanical properties of "NWC-FM" with simple concrete without adding coarse aggregate, because the filling of karst cave does not need too high strength to meet the engineering requirements.
For the configuration of filling materials for karst cave, its strength is greater than that of undisturbed soil in karst cave, which can meet the requirements of shield construction The strength of undisturbed soil in sampling and testing shall not exceed 0.5MPa. Too high strength will waste material properties and reduce economy.

8.Comments:
In table 8 and in the text, costs are expressed in yuan/t. It will be easier to compare the costs in USD.
8.Response: According to your suggestion, we have replaced the cost of RMB with USD.
